# Sesamol Attenuates Renal Inflammation and Arrests Reactive-Oxygen-Species-Mediated IL-1β Secretion via the HO-1-Induced Inhibition of the IKKα/NFκB Pathway In Vivo and In Vitro

**DOI:** 10.3390/antiox11122461

**Published:** 2022-12-14

**Authors:** Kuo-Feng Tseng, Ping-Hsuan Tsai, Jie-Sian Wang, Fang-Yu Chen, Ming-Yi Shen

**Affiliations:** 1Department of Biological Science and Technology, China Medical University, Taichung 40406, Taiwan; 2Graduate Institute of Biomedical Sciences, China Medical University, Taichung 40402, Taiwan; 3Department of Internal Medicine, Division of Nephrology, China Medical University Hospital, Taichung 40402, Taiwan; 4Department of Medical Research, China Medical University Hospital, Taichung 40402, Taiwan; 5Department of Nursing, Asia University, Taichung 41354, Taiwan

**Keywords:** renal inflammation, IKKα, heme oxygenase-1, sesamol, 5/6 nephrectomy

## Abstract

Chronic nephritis leads to irreversible renal fibrosis, ultimately leading to chronic kidney disease (CKD) and death. Macrophage infiltration and interleukin 1β (IL-1β) upregulation are involved in inflammation-mediated renal fibrosis and CKD. Sesamol (SM), which is extracted from sesame seeds, has antioxidant and anti-inflammatory properties. We aimed to explore whether SM mitigates macrophage-mediated renal inflammation and its underlying mechanisms. ApoE^–/–^ mice were subjected to 5/6 nephrectomy (5/6 Nx) with or without the oral gavage of SM for eight weeks. Blood and urine samples and all the kidney remnants were collected for analysis. Additionally, THP-1 cells were used to explore the mechanism through which SM attenuates renal inflammation. Compared with the sham group, the 5/6 Nx ApoE^–/–^ mice exhibited a significant increase in the macrophage infiltration of the kidneys (nephritis), upregulation of IL-1β, generation of reactive oxygen species, reduced creatinine clearance, and renal fibrosis. However, the administration of SM significantly alleviated these effects. SM suppressed the H_2_O_2_-induced secretion of IL-1β from the THP-1 cells via the heme oxygenase-1-induced inhibition of the IKKα-NF-κB pathway. SM attenuated renal inflammation and arrested macrophage accumulation by inhibiting IKKα, revealing a novel mechanism of the therapeutic effects of SM on renal injury and offering a potential approach to CKD treatment.

## 1. Introduction

Chronic nephritis leads to an irreversible reduction in the glomerular filtration rate and renal fibrosis, ultimately leading to chronic kidney disease (CKD) and end-stage renal disease (ESRD). The infiltration of the glomerular and interstitial macrophages is a hallmark of CKD that plays a crucial role in renal injury [1]. After kidney injury, cytokines or chemokines released by damaged cells recruit monocytes to inflammatory lesions, where they are activated and differentiated into macrophages [2]. Macrophages are crucial immunological regulators and mediators of inflammation; therefore, the secretion of macrophage-derived inflammatory cytokines, such as interleukin (IL)-1 and IL-6, is known to induce kidney damage and inflammation [3]. Inflammatory macrophages are activated and differentiated on-site, resulting in the release of IL-1, which induces the immune responses of the Th1-type cells that damage tissues [4]. In CKD, immune cells infiltrating the kidneys play a deleterious role, actively participating in the disease’s progression and leading to nephron loss and fibrosis [1,5].

The NF-κB signaling pathway is highly activated in inflammatory diseases. It is accompanied by the recruitment of inflammatory cells and the production of proinflammatory cytokines, such as IL-1, IL-6, IL-8, and tumor necrosis factor-alpha (TNF-α) [6,7]. The IκB kinase (IKK) complex, a vital component of the NF-κB signaling pathway, consists of two catalytically active kinases, IKKα and IKKβ, and the regulatory subunit IKKγ (NEMO). Notably, IKKβ reportedly induces the expression of cytokines through the phosphorylation of p65 of the NF-κB pathway [8,9]. In contrast, IKKα acts through canonical and non-canonical NF-κB pathways [10]. Moreover, IKKα reportedly mediates macrophage polarization and the secretion of fibrotic factors [11]. Several studies have examined the advantageous effect of suppressing IL-1 on the renal function [12]. Interleukin-1β (IL-1β) is released primarily from stimulated macrophages and monocytes and plays a crucial role in inflammatory and immune responses [13]. Notably, the IL-1β-dependent signaling drives renal damage and fibrosis by activating the proliferation of kidney stromal cells via the MYC transcription factor [4,14,15]. Additionally, the infiltration of macrophages in the kidneys has been associated with the level of kidney injury and kidney fibrosis in human and experimental diabetic nephropathy [5]. IL-1β contributes to renal damage; thus, its inhibition might offer a crucial way of preventing renal fibrosis.

Reactive oxygen species (ROS) are critical mediators of the activation of proinflammatory signaling pathways [16]. Therefore, the production of ROS might favor the induction of proinflammatory macrophages during CKD progression. Elevation in the levels of ROS or reactive nitrogen species, or both, is commonly observed in animal models of CKD [17,18], similar to the progression of renal disease in humans [19]. Moreover, increased intracellular levels of ROS cause the oxidation of proteins, lipids, and DNA, resulting in cellular damage in the kidneys. Hydrogen peroxide (H_2_O_2_), a major endogenous ROS, is a destructive molecule that is widely used to mimic ROS in in vitro studies [20].

Heme oxygenase-1 (HO-1) is a cellular defense mechanism activated by oxidative stress and other stimuli. It also has antioxidant and anti-inflammatory properties. Unilateral ureteral obstruction caused more severe fibrosis in HO-1 knockout mice. Furthermore, macrophage infiltration was increased in HO-1-deficient mice, as measured by the expression of F4/80 [21]. HO-1 is rapidly produced in the kidneys during glycerol-induced rhabdomyolysis, and previously, the stimulation of HO-1 with hemoglobin was shown to reduce injury [22]. The preinduction of HO-1 increases renal function and survival, whereas the pharmacological suppression of HO-1 aggravates kidney disease. HO-1 suppresses the overactive immune response in several immune and kidney-resident cells [23]. HO-1 regulates numerous pathways to mediate cytoprotection in kidney disease. Thus, the exploration of the mechanism of HO-1-mediated protection in the kidney after oxidative injury is crucial.

Sesamol (SM), 3,4-methylenedioxyphenol, is a lignan compound and an important aromatic component of sesame oil [24]. SM is present in sesame (*Sesamum indicum*) seeds and sesame oil and is continuously decomposed by sesaline and other substances during thermal processing. SM has a potent antioxidant capacity [25], as well as anti-inflammatory, antiaging, and cardioprotective potential [26,27]. SM reduced the oxidative-stress-induced lipid peroxidation and malondialdehyde (MDA) levels of ApoE^–/–^ nephrectomy mice [28].

In this study, we investigated the protective effects of SM on a 5/6 nephrectomized (5/6 Nx) mice model. In particular, we examined the influences of SM on macrophage infiltration and inflammatory mediators. Our study provides a novel mechanism of the action of SM in the prevention and treatment of renal fibrosis.

## 2. Materials and Methods

### 2.1. Cell Culture and Reagents

The Tohoku Hospital Pediatrics-1 (THP-1) cell line was purchased from the American Type Culture Collection (Rockville, MD, USA). THP-1 cells, a human leukemia monocytic cell line, were maintained in RPMI-1640 medium (Gibco BRL, Gaithersburg, MD, USA) supplemented with 25 U mL^−1^ penicillin and streptomycin and 10% fetal bovine serum (HyClone Laboratories, Logan, UT, USA). Sesamol SM (5-hydroxy-1,3-benzodioxole; purity: 98%), dimethyl sulfoxide (DMSO), phorbol 12-myristate 13-acetate (PMA), and H_2_O_2_ solution were purchased from Sigma (St. Louis, MO, USA). IKKα siRNA (si-IKKα) and control siRNA (si-Ctl) were procured from Santa Cruz Biotechnology (Santa Cruz, CA, USA).

### 2.2. Animal Design

All the mouse experiments were approved by the Institutional Animal Care and Use Committee of China Medical University and performed in accordance with the National Institutes of Health Laboratory Animal Care and Use Guidelines (NIH Publication No. 85-23, revised edition 1996). Female ApoE^–/–^ mice (Jackson Laboratory, Sacramento, CA, USA) aged 10–12 weeks (weighing 21–23 g) underwent 5/6 nephrectomy (5/6 Nx), the complete removal of the right kidney, and the infarction of approximately 2/3rd of the left kidney by ligating part of the renal artery branch. Female mice were used owing to their small glomerular sieving structure [29], which is beneficial for the development of CKD. Moreover, lipid abnormalities often accompany and aggravate renal disease. ApoE^–/–^ mice, as a dyslipidemia animal model, that underwent 5/6Nx easily accumulated a greater number of immune cells in the kidneys, making them ideal for studying kidney inflammation. Throughout the study, the mice were fed on a regular diet. The mice received SM (25 or 50 mg/kg) via oral gavage thrice per week for eight weeks after 5/6 Nx surgery. All the mice were euthanized by isoflurane (Abbott Laboratories, Chicago, IL, USA) inhalation.

### 2.3. Plasma Biochemical Analyses

We separated the plasma from the whole blood and then removed the platelets from the plasma by centrifugation (centrifuge 5415 R, Eppendorf). The test paper and plasma (150 μL) were then placed in an automatic clinical chemistry analyzer (Arkray SPOTCHEM EZ SP-4430, Kyoto, Japan) according to the manufacturer’s instructions.

### 2.4. Histology Staining, Immunohistochemistry, and Masson’s Trichrome Staining

The kidney tissues were embedded in paraffin and sliced into 3 µm-thick sections. The slides were stained with hematoxylin and eosin (H&E, Abcam, Cambridge, UK), antibodies, or Masson’s trichrome stain (Sigma). Images were captured using a DM750 microscope (Leica, Wetzlar, Germany) and analyzed using the ImageJ software (version 1.52a, NIH, Bethesda, MD, USA).

### 2.5. Cell Viability Assay

For this assay, 1 × 10^4^ THP-1 cells were inoculated on 96-well plates for the cell viability assessment. PMA was added for 48 h to render the cells adherent, after which the medium was changed to a fresh medium for 24 h. The cells were treated with H_2_O_2_ (0–30 µM) or SM (0–20 µM) for 24 h. Subsequently, the cells were incubated with 10% WST-1 reagent (Abcam) in a fresh medium at 37 °C for 4 h. The absorbance of each sample at 440 nm was measured using a microplate reader (Infinite M1000, TECAN, Mechelen, Belgium). The percentage of each sample was calculated after normalizing all the data to those of the corresponding controls.

### 2.6. Intracellular Reactive Oxidative Stress Measurement

The intracellular levels of ROS were determined using a cellular ROS/superoxide detection assay kit (Abcam) according to the manufacturer’s instructions. The cells (1 × 10^4^) were inoculated on 96-well plates, and PMA was added to render cells adherent for 48 h, after which the medium was changed to fresh medium for 24 h. The THP-1 cells were treated with H_2_O_2_ (0–30 µM) for 24 h. Subsequently, the supernatant was removed, and the cells were washed with phosphate-buffered saline (PBS). The 2′,7′-dichlorofluorescein diacetate (DCFH-DA) ROS-specific stain was then added, and the cells were incubated in the dark at 37 °C for 30 min. The cells were then washed twice with PBS. The intracellular levels of ROS were determined using an Infinite M1000 microtiter plate reader (TECAN) (Ex = 488 nm, Em = 520 nm).

### 2.7. Immunoblot Analyses

THP-1 cells were seeded on six-well plates at a density of 1 × 10^7^ cells per well and processed as previously described. The cells were then treated with vehicle, H_2_O_2_ (5 µM for 24 h) alone, or H_2_O_2_ after an SM (2 µM for 1 h) pretreatment. To obtain protein extracts, the cells were lysed and homogenized in RIPA lysis buffer in the presence of protease inhibitors (Roche Applied Science, Penzberg, Germany). After centrifugation, the protein concentration was determined using a BSA protein assay kit (Pierce Biotechnology Inc., Waltham, MA, USA). For the immunoblot assays, 20 µg of cell lysate was loaded onto 12% SDS-PAGE gel per well and separated by electrophoresis. The separated proteins were transferred to Hybond-PVDF membranes (GE Healthcare Amersham, Buckinghamshire, UK). The membranes were blocked with blocking buffer for 1 h at 25 °C, incubated overnight at 4 °C with primary antibodies (1:1000), and finally incubated with appropriate secondary antibodies (1:5000). The primary antibodies used included anti-p-IKKα (1:1000, #44-714) and anti-IKKα (1:1000, #14A231) (Invitrogen, Waltham, MA, USA), anti-p-NF-кB (1:1000, #3033) and anti-p-IкBα (1:1000, #9246) (Cell Signaling Technology, Danvers, MA, USA), anti-HO-1 (1:1000, ab13243, Abcam), and anti-β-actin (1:10,000, A5441, Sigma). β-actin was used as the loading control. The protein expression was evaluated using an ECL reagent (Millipore, Billerica, MA, USA). Images were obtained using quantitative video densitometry (G-box Image System; Syngene, Frederick, MD, USA).

### 2.8. Transfection of siRNA

The cells were inoculated on six-well plates at approximately 5 × 10^6^ cells per well and stimulated with PMA for 48 h. The medium was replaced with fresh medium supplemented with 20% FBS without antibiotics. The control and IKKα-siRNA (Santa Cruz Biotechnology, Santa Cruz, CA, USA) were prepared at a final concentration of 10 nM, following the manufacturer’s instructions. The control and IKKα-siRNA and the siRNA transfection reagent (INTERFERin^®^, Polyplus-transfection, Illkirch, France) were mixed for 10 min and then transfected into the cells for 12 h. H_2_O_2_ was then added to stimulate the cells.

### 2.9. Quantitative Real-Time PCR

THP-1 cells were seeded at a density of 1 × 10^6^ cells/well on 6-well plates and treated with different stimuli. For the in vivo experiments, the murine renal tissues were homogenized. All the samples were collected and frozen with NucleoZOL (Macherey-Nagel, Düren, Germany) at −80 °C, and total RNA was isolated and extracted by centrifugation according to the manufacturer’s instructions. The iScript cDNA Synthesis Kit (Bio-Rad, Hercules, CA, USA) was used to synthesize the cDNA. Finally, the Q SYBR Green Supermix (Bio-Rad) was used for the real-time PCR. The primers used are listed in Table 1.

### 2.10. Docking Pose Prediction

The crystal structures of IKKα (PDB code: 5EBZ, deposited date: 20 October 2015 and HO-1 (PDB code: 1N3U, deposited date: 29 October 2002) were obtained from the Protein Data Bank (PDB, https://www.rcsb.org/) for the first step of the docking simulation. The docking of IKKα and HO-1 with the protein catalytic sites was evaluated using BIOVIA Discovery Studio (Dassault Systèmes, Vélizy-Villacoublay, France).

### 2.11. Statistical Analysis

The results are presented as the mean ± standard error of the mean (SEM) for each repeated experiment. Statistical significance (*p* < 0.05) was assessed by one-way ANOVA for multiple groups and Student’s *t*-test using the GraphPad Prism 5 software (GraphPad Software Inc., La Jolla, CA, USA).

## 3. Results

### 3.1. Sesamol Ameliorated Renal Inflammation and Fibrosis in 5/6 Nx ApoE^−/−^ Mice

We performed 5/6 nephrectomy (5/6 Nx) on the ApoE^−/−^ mice and established a CKD mouse model (Figure 1A). In addition, we photographed the kidneys of the mice in all the groups (Figure 1B). After euthanasia, we collected samples of urine and analyzed the levels of creatinine in comparison with those of serum creatinine to calculate the creatinine clearance rate (CCR), which is an indicator of renal function (Figure 1C–E). We observed a decrease in the CCR in the 5/6 Nx group, indicating renal failure. In particular, we demonstrated that the 5/6 Nx mice exhibited higher serum creatinine and lower urine creatinine levels than the control, suggesting compromised creatinine clearance. However, we observed that the SM administration alleviated this reduction in the CCR. The nucleated cell count suggested the aggravation of inflammation in the 5/6 Nx mice (Figure 1F, *p* < 0.01). We observed that SM reduced inflammation, with a decrease in the number of nucleated cells. The representative optical microscopy images showed glomerular lipidosis, which was characterized by dilated glomerular capillary loops containing foam cells and cholesterol accumulation in the 5/6 Nx group (Figure 1F). In addition, the micrographs of the renal tissues of the 5/6 Nx group showed enlarged glomeruli and apoptotic proximal tubules, resulting in the vacuolation of the renal tubular epithelial cells. However, we observed that these effects were alleviated after treatment with SM (Figure 1F). Masson’s trichrome staining of the kidney tissues of the 5/6 Nx mice showed a severe deposition of collagen, which is related to fibrosis (Figure 1G, *p* < 0.001). Furthermore, we observed that the mice in the SM group exhibited an enhanced reduction in the Masson’s trichrome-stained positive area compared with that of the 5/6 Nx mice (Figure 1G). These results suggested that the 5/6 Nx mice were affected by immune-response-related nephritis. The administration of SM inhibited inflammation and improved renal function. These findings imply that SM reduced fibrosis in the mice with CKD by inhibiting collagen deposition.

### 3.2. SM Ameliorated Macrophage Infiltration and Reduced the Levels of P-IKKα and IL-1β and the Generation of ROS in the 5/6 Nx ApoE^−/−^ Mice

To emphasize the relevance of the macrophages in CKD, we assessed the expression of F4/80-positive (F4/80+) macrophages in the kidneys and quantified the number of infiltrating F4/80+ macrophages. We observed that 5/6 Nx led to strong F4/80+ macrophage infiltration in the kidneys (Figure 2A, *p* < 0.001). In contrast, the number of macrophages was significantly reduced in the kidneys of the mice to which SM was administered compared with that of the mice that received 5/6 Nx alone (Figure 2A, *p* < 0.05). This showed that SM resulted in a lowered infiltration of macrophages in the kidneys. Moreover, we tested the ability of P-IKKα to induce the infiltration of macrophages into inflamed tissues. We observed that the expression of IKKα was significantly increased in the 5/6 Nx group compared with that in the sham group (Figure 2B, *p* < 0.001). However, it was decreased in the 5/6 Nx group treated with SM. In addition, we also calculated the expression level of IL-1β. We observed that IL-1β expression was increased in the 5/6 Nx group, whereas it was reduced in the SM-treated group (Figure 2C).

Additionally, we measured the levels of renal malondialdehyde (MDA), an indicator of lipid peroxidation, to confirm the observed levels of ROS [28]. We found that the levels of MDA were higher in the 5/6 Nx group compared with those in the sham group (Figure 2D, *p* < 0.01). Notably, SM ameliorated the increase in the levels of renal MDA in the 5/6 Nx ApoE^−/−^ mice (Figure 2D). Moreover, we found that the reduced levels of renal glutathione peroxidase (GPx) were attributed to the prolonged exposure to elevated oxidative stress in the kidney in the 5/6 Nx group (Figure 2E, *p* < 0.01). In contrast, the levels of GPx were increased in the SM group (Figure 2E). These data suggested that the infiltration of macrophages and increased formation of foam cells were associated with higher levels of ROS and that the phosphorylation of IKKα led to the secretion of IL-1β in the kidneys of the 5/6 Nx ApoE^−/−^ mice, which were inhibited by the administration of SM.

### 3.3. SM Improved the Blood Lipid Profile and Plasma MDA Levels in the 5/6 Nx ApoE^−/−^ Mice

Abnormal levels of blood lipids and lipid peroxidation have been shown to play a significant role in the progression of inflammation-mediated CKD [30]. Therefore, we further explored the effects of SM on the blood lipid profiles and levels of plasma MDA in the 5/6 Nx ApoE^−/−^ mice. After euthanasia, the whole blood was separated to obtain plasma, and we performed biochemical analyses (Figure 3A–E). We observed that the levels of triglycerides, total cholesterol, and blood glucose were increased after eight weeks in the 5/6 Nx mice (Figure 3A,B,E, *p* < 0.05). Furthermore, we detected a downward trend in the high-density lipoprotein cholesterol (HDL-C) levels (Figure 3C). In contrast, the level of low-density lipoprotein cholesterol (LDL-C) was significantly increased (Figure 3D, *p* < 0.05) in the 5/6 Nx mice compared with that of the sham mice. However, SM reduced the levels of blood triglyceride, total cholesterol, and LDL-C (Figure 3A,B,D, *p* < 0.05) and increased those of HDL-C (Figure 3C, *p* < 0.05) in the 5/6 Nx mice. We further noticed that the blood glucose level was significantly decreased in the high-dose SM-treated group (Figure 3E, SM50, *p* < 0.05). In addition, compared with the sham group, the 5/6 Nx group exhibited increased levels of plasma MDA (Figure 3F, *p* < 0.05). In contrast, the levels of plasma MDA in the SM groups were lower than those in the 5/6 Nx group. These results implied that uncontrolled levels of blood lipids result in high levels of ROS during the progression of chronic renal disease that can be conversely blocked by SM.

### 3.4. IKKα Mediated Cytokine IL-1β Secretion from the Macrophages and Promoted Inflammation

To verify whether ROS directly influence the macrophages, we treated phorbol-myristate-acetate (PMA)-induced THP-1 cells with H_2_O_2_. The THP-1 cells were differentiated into macrophages using 25 ng/mL PMA for two days and fresh medium for one day. The WST-1 assay was conducted to examine the viability of the H_2_O_2_-injured THP-1 cells. The data showed that H_2_O_2_ reduced the THP-1 cells’ activity in a concentration-dependent manner, and the half maximal inhibitory concentration of H_2_O_2_ was about 5 μM (Figure 4A, *p* < 0.01). Next, we observed that H_2_O_2_ induced the release of intracellular ROS in the THP-1 cells (Figure 4B, *p* < 0.05). In addition, we demonstrated that the ROS induced the expression of IL-1β in the H_2_O_2_-primed THP-1 cells in a concentration-dependent manner (Figure 4C, *p* < 0.05). After confirming the dosage of H_2_O_2_, we used a dose of 5 µM H_2_O_2_ for the following experiments. Next, we assessed the role of IKKα in the ROS-induced secretion of IL-1β in the macrophages. We administered the siRNA-targeting scrambled sequence or IKKα (siIKKα) to the cells for 12 h and then stimulated them with H_2_O_2_ for another 24 h. We added BMS345541 (an IKKα and IKKβ inhibitor) 1 h prior to H_2_O_2_, which served as a positive control. We noticed that the stimulation of the THP-1 cells with H_2_O_2_ increased the expression of IL-1β, whereas the knockdown of IKKα by siIKKα or its pharmacological inhibition by BMS345541 decreased the expression of IL-1β in the THP-1 cells (Figure 4D, *p* < 0.05). In contrast, siCtrl blockade did not affect the H_2_O_2_-stimulated THP-1 cells. These findings indicated that IKKα plays a pivotal role in the expression of IL-1β in THP-1 cells. We also observed that the THP-1 cells exhibited the phosphorylation of IKKα, IкBα, and NF-кB (p65) upon their stimulation with H_2_O_2_ (Figure 4E). Notably, siIKKα suppressed IKKα and the phosphorylation of IKKα, IкBα, and NF-кB (p65) in the THP-1 cells (Figure 4E,F, *p* < 0.05). We observed that most of the remaining THP-1 cells after siIKKα blockade had a lower expression of IL-1β, suggesting that reduced cellular inflammatory events were associated with IKKα blockade. Therefore, the inhibition of IKKα might represent a promising anti-inflammatory treatment strategy.

### 3.5. SM Increased the Level of HO-1 to Inhibit IKKα and Blocked the Secretion of IL-1β in the Macrophages

We evaluated the effect of SM on the H_2_O_2_-stimulated THP-1 cells. We administered SM for 1 h and then stimulated the cells with H_2_O_2_ for the following 24 h. We did not observe any SM-induced cytotoxicity at a concentration of 20 μM in the WST-1 assay (Figure 5A, *p* < 0.05). In addition, the expression of IL-1β was significantly inhibited by SM compared with that in the H_2_O_2_-treated THP-1 cells (Figure 5B, *p* < 0.01). We also observed higher phosphorylation levels of IKKα, IкBα, and p65 in the H_2_O_2_-treated THP-1 cells. In contrast to the H_2_O_2_ treatment, the administration of SM reduced the phosphorylation of IKKα, resulting in lower levels of p-IкBα and P-p65 (Figure 5C,D). These data showed that SM blocked the expression of IL-1β in the THP-1 cells. Moreover, we speculated that SM would increase the heme oxygenase-1 (HO-1) levels, thereby inhibiting IKKα phosphorylation (Figure 5E). Notably, HO-1 inhibits oxidative damage and inflammation, significantly reducing the occurrence of inflammatory events [31]. Therefore, we assessed the levels of HO-1 in the THP-1 cells to investigate the anti-inflammatory response. We found that the administration of SM increased the transcription of HO-1 (Figure 5F, *p* < 0.01). Moreover, when the macrophages were exposed to H_2_O_2_, they displayed a significant inflammatory response, as evidenced by their expression of IL-1β. In particular, we found that the expression of IL-1β was substantially reduced in the presence of SM (Figure 5G, *p* < 0.05). In contrast, SnPP (tin protoporphyrin IX dichloride, an HO-1 inhibitor) abolished the anti-inflammatory effect of SM on the H_2_O_2_-induced THP-1 cells. To test whether HO-1 has the potential to inhibit IKKα-mediated inflammation, we performed molecular docking simulations to predict the activation and binding affinities of HO-1 in IKKα inhibition (Figure 5H, Appendix A). We observed that IKKα, being involved in the inflammatory response, interacted with HO-1, and the energy value of the IKKα–HO-1 interaction was −0.62 kCal/mol. In the H_2_O_2_-treated THP-1 cells, the expression of p-IKKα and IKKα was increased, whereas that of HO-1 was decreased. Conversely, after the addition of SM, the expression of HO-1 was increased, resulting in a decrease in the expression of IKKα (Figure 5I). We obtained similar results in vivo (Figure 5J). In conclusion, HO-1 might counteract the underlying mechanism of the ROS-induced inflammatory response by interfering with the activation of IKKα.

## 4. Discussion

Renal inflammation is an initial response to renal injury. Inflammation promotes the process of fibrosis, ultimately leading to CKD and ESRD. However, we lack effective strategies that can be used to improve this condition. SM is a natural organic compound extracted from sesame seeds with antioxidant and anti-inflammatory properties. Notably, 5/6 nephrectomy (5/6 Nx) in mice mimics renal failure after the loss of kidney function in humans, exacerbates renal injury through inflammation and oxidative stress, and has been widely used in CKD research [28]. In this study, we demonstrated that SM ameliorated renal inflammation and fibrosis in 5/6 Nx ApoE^−/−^ mice. Moreover, we also observed that through the IKKα pathway, renal ROS induce the activation of macrophages and the secretion of IL-1β, which recruits more immune cells (nucleated cells, including macrophages) to the renal tissues, leading to the increased formation of foam cells and increased production of ROS, which cause inflammation and fibrosis in the kidneys, resulting in CKD. However, SM was shown to arrest this process via an increase in the levels of HO-1, which led to the inhibition of IKKα in the macrophages (Figure 6).

Inflammation is closely related to renal disease and can be defined as a complex network of interactions between renal parenchymal cells and resident immune cells, including macrophages, dendritic cells, circulating monocytes, lymphocytes, and neutrophils. The kidney harbors various resident immune cells that play crucial roles in maintaining tissue homeostasis. These cells produce inflammatory mediators that initiate kidney disease and concomitantly trigger a regulatory response to curb inflammation, repair damaged tissue, and restore homeostasis. Inflammation plays a vital role in the progression of chronic renal failure. Excessive inflammation activates the white blood cells, leading to the production of cytokines related to splenomegaly. The inflammatory responses in each group of mice were compared by calculating the size of the spleen (data not shown). In this study, we observed that nucleated cells infiltrated the kidneys of the 5/6 Nx ApoE^−/−^ mice, and this infiltration was ameliorated by SM.

Renal fibrosis is the final critical common pathway in the progression of renal diseases. Renal fibrosis has been associated with inflammatory cell infiltration, tubular atrophy, and the loss of peritubular or glomerular capillaries, during which a sustained inflammatory process might result in substantial tissue damage, eventually leading to fibrogenesis [1,12]. Therefore, anti-inflammatory therapies should be investigated for the treatment of renal fibrosis. Acute kidney injury (AKI) and CKD are associated with increased numbers and infiltration of the macrophages in the kidney [1,3]. Macrophages are a heterogeneous population of cells with diverse functions and phenotypic plasticity. They are also known to play pathogenic roles in renal inflammation and fibrosis. A recent study also supported the notion that macrophages directly contribute to the development of fibrosis by sustaining the renal inflammatory milieu [5]. In addition, regardless of the etiology, renal parenchymal injury induces the release of chemotactic factors, renal infiltration of macrophages, and polarization to the M1 phenotype. M1 macrophages are characterized by the production of inflammatory cytokines, which can lead to tubular cell death, endothelial dysfunction, inflammation, and fibrosis, while suppressing M2 repolarization. Thus, the sustained infiltration of M1 macrophages might contribute to the progression of renal injury [32]. The causal role of the classically activated macrophages in nephritis has been demonstrated by studies showing that macrophage depletion or the inhibition of monocyte chemoattractant proteins reduced the number of renal macrophages and protected the kidney [5]. Inhibiting the infiltration of macrophages reduces proteinuria and structural damage to the kidney [5,12]. Thus, the inhibition of the macrophages is likely to be a beneficial approach for the treatment of CKD. In this study, we found that SM inhibited macrophage infiltration and the ROS-induced expression of IL-1β in the kidneys of ApoE^−/−^ 5/6 Nx mice.

CKD is associated with dyslipidemia, including high levels of triglycerides, low levels of HDL cholesterol, and an altered lipoprotein composition [33]. Increasing evidence supports the notion that lipids are nephrotoxic [29,34]. The elevated levels of some lipids result in increased inflammation and oxidative stress in CKD. A high-fat diet causes lipid accumulation, as well as structural and functional abnormalities in the proximal tubular epithelial cells and the release of large amounts of proinflammatory cytokines [35]. In our pilot study, we observed changes in the glomerular cluster structure and glomerular microvascular lipid accumulation, as indicated by the H&E staining and Oil Red O staining of the kidneys of the 5/6 Nx ApoE^−/−^ mice. This situation was improved after the oral administration of SM. Moreover, we assessed the levels of MDA to analyze the production and levels of ROS. The levels of ROS were increased in the ApoE^−/−^ mice with 5/6 Nx, which presented higher levels of triglycerides, total cholesterol, and LDL-c in the plasma and a higher inflammatory response in the kidneys. Regarding the reduction in glomerular lipid deposition in the 5/6 Nx mice after the SM treatment, we found that SM protected the tissue against ROS damage and improved glomerular damage attributed to ROS as the main inflammatory factor. Lipids promote the recruitment of immune cells, such as macrophages, neutrophils, and bone-marrow-derived dendritic cells, to the tissues. In patients with kidney disease, the inflammatory response was caused by an increase in the levels of triglyceride-rich lipoproteins, and patients on hemodialysis had significantly lower inflammation and levels of lipoprotein variables than patients with CKD [36]. These findings implied that the beneficial effects of lowering the levels of triglyceride-rich lipoproteins might reduce the inflammatory response.

Macrophages, which are monocyte-derived phagocytic cells, are located in peripheral tissues and act as important mediators of inflammation and immune modulation [2]. They are prevalent in the kidneys of humans with chronic renal disease. Classically activated macrophages release inflammatory cytokines and promote oxidative stress, leading to the development of renal fibrosis [5,32]. Although the macrophages play a crucial role in scavenging cellular debris, the increased renal infiltration of macrophages can tip the balance, thus causing local injury and inflammation. Injury and inflammation are mediated by the release of macrophage-derived inflammatory cytokines, such as interleukin (IL)-1, IL-6, and IL-23, and the generation of reactive oxygen/nitrogen species, each of which has been implicated in impaired renal function [1,3]. Generally, cytokines contribute to the pathogenesis of kidney disease by upregulating endothelial cell adhesion molecules and chemokines that further promote renal immune cell infiltration [3]. In addition, many cytokine-activated signaling pathways increase the activation of IKK and NF-κB, further promoting a proinflammatory phenotype [37]. Evidence suggesting that the NF-ĸB system is involved in the development of renal inflammation and injury has been described in studies of various clinical and experimental renal diseases. The expression or activation of NF-κB is increased in the kidneys of patients with glomerulonephritis, diabetic nephropathy, and AKI [37]. One of the consequences of this inflammatory cycle is the promotion of local oxidative stress, which enhances renal injury and impairs the renal tubular and hemodynamic functions. In this study, we identified IKKα as a regulator of inflammation in kidney diseases. The IKK complex is essential for the activation of NF-кB during inflammation and oxidative stress stimulation. The NF-κB pathway consists of canonical and non-canonical pathways. The canonical NF-κB (p65/p50) pathway is activated by various stimuli, transducing a quick but transient transcriptional activity, to regulate the expression of various proinflammatory genes and serve as the critical mediator of the inflammatory response. The non-canonical NF-κB pathway is also an important arm of NF-κB signaling that predominantly targets the activation of the p52/RelB NF-κB complex. However, canonical NF-κB responses are much faster than non-canonical NF-κB pathway signaling, making this pathway particularly important for the innate immune response of first responder cells, such as the macrophages [6,7]. NF-кB is involved in the regulation of proinflammatory signals [37]. The IKKβ subunit is a major activator of NF-кB and the underlying inflammatory response. Therefore, we used siIKKα to demonstrate that IKKα might induce the activation of NF-кB. Several studies support our findings [38,39]. Moreover, we observed that the stimulation of THP-1 cells with H_2_O_2_, which increased the expression of IL-1β, promoted inflammation. Conversely, this effect was inhibited by incubating the THP-1 cells with siIKKα or BMS345541 (an IKK inhibitor). Furthermore, considering that the inhibition of IKKα by siIKKα might affect the transcription and translation of IKKβ, we examined the protein expression of IKKβ and found that it was not affected. In addition, we also found that p-IKKα and the total IKKα were both elevated in the H_2_O_2_-induced cells. Another article also found similar results [40]. These results indicated that H_2_O_2_ induced the expression of IL-1β in the THP-1 cells via the IKKα/NF-кB signaling.

As mentioned previously, research on the benefits of exploiting drug mechanisms in kidney disease is still lacking. In this study, we found that SM diminished the expression of IL-1β by inducing the expression of HO-1 and inhibiting the IKKα pathway. Furthermore, we confirmed the inhibitory effect of HO-1 on the IKK pathway by assessing the protein-binding site. In addition, although we did not explore fibrosis-related proteins, the findings of Masson’s trichrome staining were consistent with some studies reporting a synergistic role of IL-1β in renal fibrosis. Therefore, we concluded that sesame oil and nutritional supplements containing SM might be used to reduce the burden on the kidney function. In view of its safety, low price, and wide availability, SM can serve as a potential therapeutic agent for the treatment of kidney disease. However, more studies are required to determine the efficacy and safety of SM for patients with CKD.

This study had some limitations. Inflammation and lipids play significant roles in the progression of CKD. Macrophages take up lipids and readily form foam cells. The ApoE^−/−^ mice that underwent 5/6 Nx and did not receive the SM treatment accumulated a greater number of immune cells in the kidneys, indicating the formation and transportation of an increased number of foam cells in the blood, thus promoting glomerular lipid deposition. Foam cells are encountered in common glomerular diseases, such as focal and segmental glomerulosclerosis and diabetic nephropathy, but their pathogenic significance remains unknown. Extrapolating from studies on atherosclerosis, we can observe that therapeutics targeting the production of mitochondrial ROS or modulating cholesterol and lipoprotein uptake or their egress from these cells might prove beneficial for the treatment of kidney diseases in which foam cells are present. Secondly, different macrophages reflect different physiological roles in kidney disease. Thus, further understanding of the influences of specific subpopulations of macrophages will be helpful for the development of treatment strategies for kidney diseases. Furthermore, several studies have analyzed specific markers of each immune cell, which is a more accurate measurement method. However, this was another limitation of our study. Therefore, in future studies, the immune system should be thoroughly investigated to clarify the role of each component in kidney diseases. Thirdly, HO-2 is a factor affecting macrophage polarization and phagocytic activity in inflammation. Namely, HO-2 deficiency impairs HO-1 expression and inducibility in the macrophages [41]. In other research, HO-2 deficiency was found to enhance STZ-induced renal dysfunction and morphological damage [42]. However, HO-2 expression may reduce the effects of lipid peroxidation in the kidney [43]. Thus, our future studies may focus on the expression of HO-2 and its underlying mechanism. Fourthly, although structural and functional problems affecting the kidney are the leading cause of CKD, the current therapeutic strategies target the blood pressure, blood sugar, and dyslipidemia, which do not address fundamental etiologies, such as inflammation and fibrosis. Although kidney transplantation is the best option for enhancing the kidney function, there are critical limitations affecting the identification of a suitable donor. Thus, developing novel therapeutic strategies for the treatment of CKD is crucial for addressing the structural problems and dysfunction of the damaged kidney. Sesamol, which is derived from sesame seeds, is a natural traditional health food that has been used in ancient medicine for a long time. Many studies point to the protective effect of sesamol on kidney disease. Therefore, SM is highly suitable as a treatment or nutritional supplement for CKD patients.

## 5. Conclusions

In summary, our results highlighted that IKKα is a central regulator of the macrophages during inflammation in CKD. In addition, we demonstrated that SM treatment increased the levels of HO-1 and inhibited the ROS-induced IKKα signaling in the macrophages during inflammation in vitro and in vivo and, thus, might be beneficial for patients with CKD.

## Figures and Tables

**Figure 1 antioxidants-11-02461-f001:**
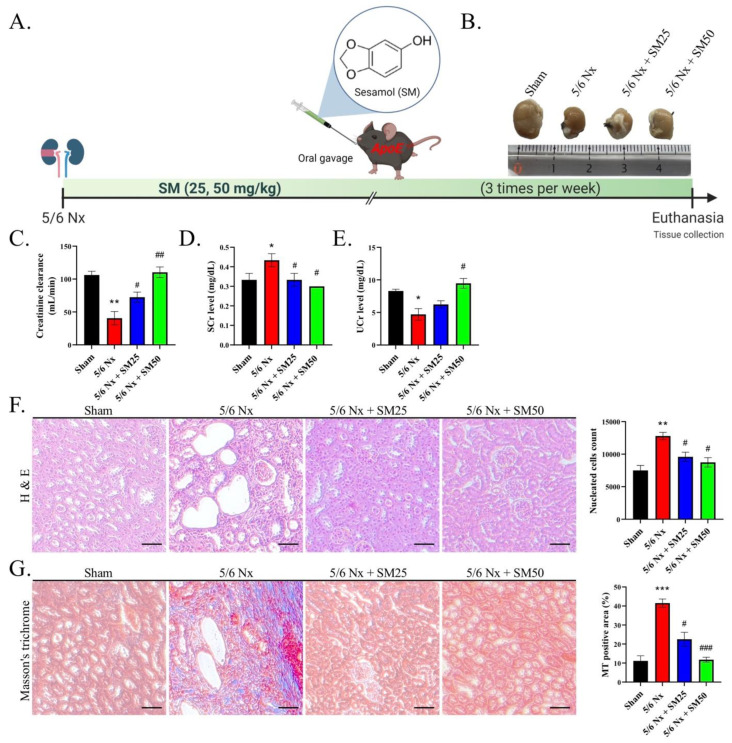
Evaluation of renal function and histopathology after eight weeks of 5/6 Nx. (**A**) Schematic representation of the 5/6 nephrectomy model of the ApoE^−/−^ mice (*n* = 6 per group). (**B**) Representative images of the kidneys from each group. (**C**) Creatinine clearance, (**D**) serum creatinine (Scr), and (**E**) urine creatinine (Ucr). Creatinine clearance was evaluated by collecting urine after 24 h. The urine and serum creatinine levels in the mice were measured after different treatments. (**F**) Observation and calculation of changes in the glomerular tuft architecture, glomerular microvascular lipid accumulation, and infiltration of immune cells (nucleated cells) in the kidneys (H&E staining) (200×); scale bar = 100 μm. (**G**) Images of kidney tissues with Masson’s trichrome staining. Masson’s trichrome-stained positive area; scale bar = 100 μm. Data are expressed as the mean ± SEM (*n* = 6). * *p* < 0.05, ** *p* < 0.01, *** *p* < 0.001 vs. sham group. # *p* < 0.05, ## *p* < 0.01, ### *p* < 0.001 vs. 5/6 Nx group.

**Figure 2 antioxidants-11-02461-f002:**
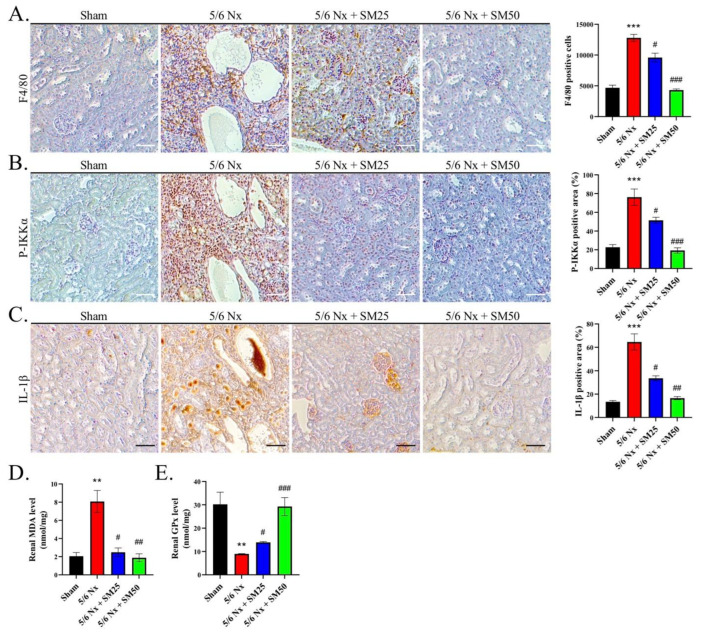
SM reduces the infiltration of macrophages and inhibits the ROS-induced secretion of IL-1β in the kidneys of ApoE^−/−^ 5/6 Nx mice. (**A**) Expression of F4/80 macrophages, (**B**) P-IKKα expression, and (**C**) IL-1β expression, as indicated by immunohistochemistry (IHC) staining; scale bar = 100 μm. Quantification of renal (**D**) MDA and (**E**) GPx levels in the kidneys of 5/6 Nx mice. Data are expressed as the mean ± SEM (*n =* 6). ** *p* < 0.01, *** *p* < 0.001 vs. sham group. # *p* < 0.05, ## *p* < 0.01, ### *p* < 0.001 vs. 5/6 Nx group.

**Figure 3 antioxidants-11-02461-f003:**
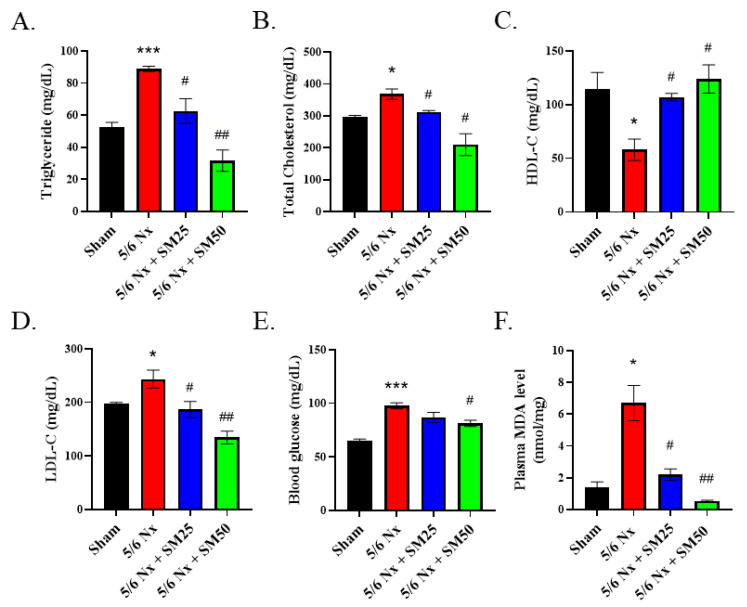
Biochemical analyses of the plasma of female ApoE^–/–^ mice subjected to different treatments. Levels of (**A**) triglycerides, (**B**) total cholesterol, (**C**) HDL-cholesterol, (**D**) LDL-cholesterol, (**E**) blood glucose, and (**F**) plasma MDA. Data are expressed as the mean ± SEM (*n* = 6). * *p* < 0.05, *** *p* < 0.001 vs. sham group. # *p* < 0.05, ## *p* < 0.01 vs. 5/6 Nx group.

**Figure 4 antioxidants-11-02461-f004:**
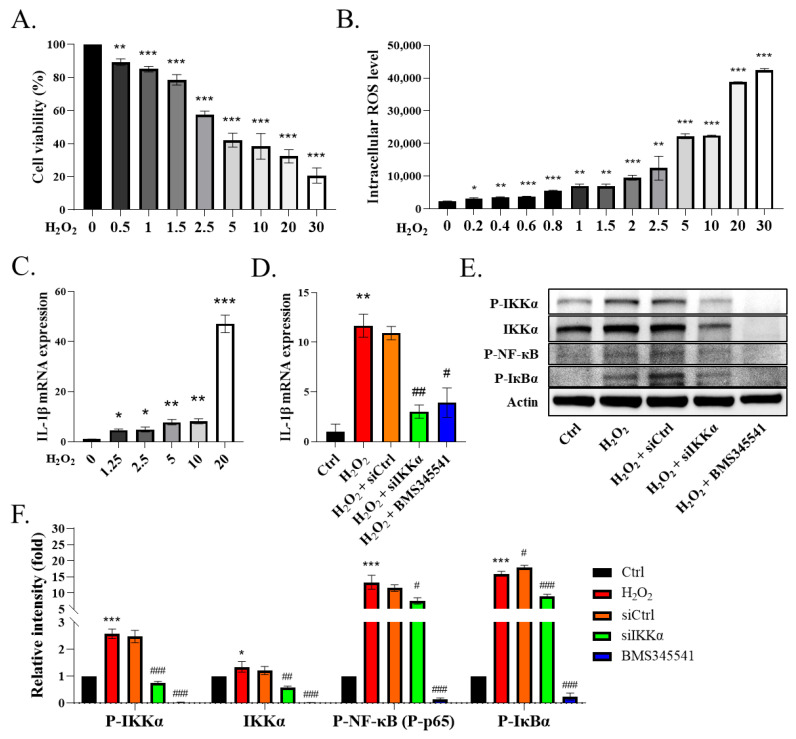
Knockdown of IKKα decreases NF-κB/IL-1β signaling in THP-1 cells. (**A**) Viability of H_2_O_2_ (0–30 μM)-treated THP-1 cells based on the WST-1 assay. (**B**) Levels of ROS measured by DCFH-DA staining. (**C**) mRNA levels of IL-1β in H_2_O_2_ (0–20 μM)-treated THP-1 cells. (**D**) mRNA level of IL-1β. THP-1 cells were transfected with siRNA-targeting IKKα (siIKKα) or scrambled sequence (siCtrl) for 12 h. BMS-345541 was used to block IKKα/β. The cells were then stimulated (or not) with H_2_O_2_ (5 μM) for 24 h. The cells were harvested for qRT-PCR assay or (**E**) immunoblotting. (**F**) Quantification of immunoblotting. Data are expressed as the mean ± SEM (*n* = 4). * *p* < 0.05, ** *p* < 0.01, *** *p* < 0.001 vs. control or 0 μM group. # *p* < 0.05, ## *p* < 0.01, ### *p* < 0.001 vs. H_2_O_2_ group.

**Figure 5 antioxidants-11-02461-f005:**
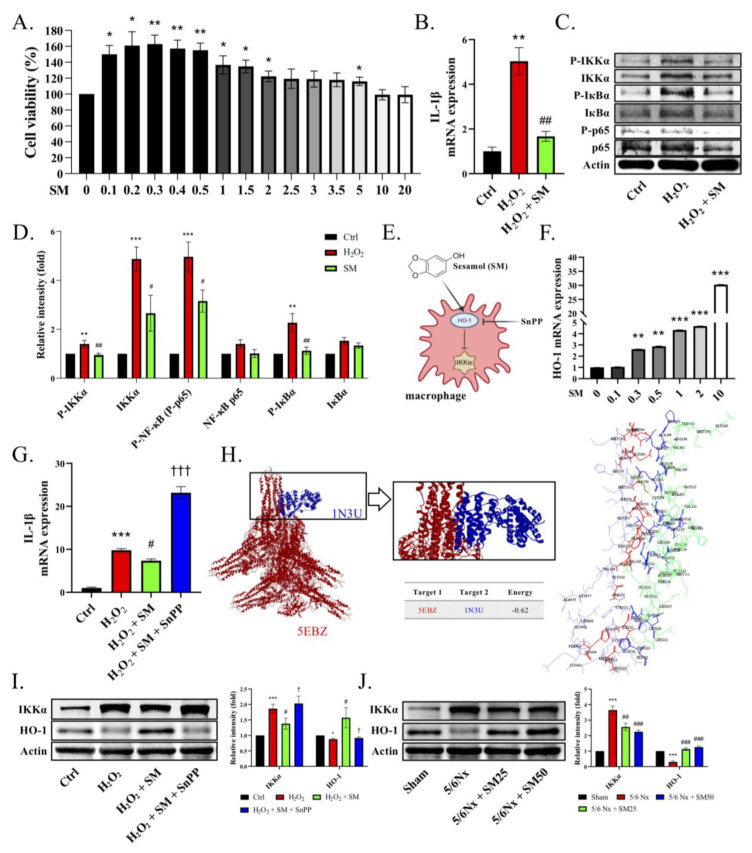
SM inhibits the IKKα signaling pathway via the upregulation of HO-1 in THP-1 cells. The THP-1 cells were differentiated using 25 ng/mL PMA for two days and fresh medium for one day. (**A**) Viability of SM-treated THP-1 cells measured by WST-1 assay. (**B**–**D**) The THP-1 cells were pretreated with SM or DMSO for 1 h and then stimulated with H_2_O_2_ (5 μM) for 24 h. The cells were harvested for qRT-PCR or immunoblotting. (**B**) Transcriptional levels of IL-1β or (**C**,**D**) immunoblotting of p-IKKα, IKKα, p-p65, and p-IкBα. (**E**) Expected interactions of SM, heme oxygenase-1 (HO-1), and IKKα. (**F**) Transcriptional levels of HO-1 after treatment with SM. (**G**) Transcriptional levels of IL-1β. (**H**) Visualization of the molecular docking pose for the interaction between IKKα and HO-1. Protein–protein interaction between IKKα (PDB code: 5EBZ; red structure) and HO-1 (PDB code: 1N3U; blue structure) in 3D. Protein–protein interaction residue. (**I**) Immunoblotting of IKKα and HO-1 after the administration of H_2_O_2_, SM, and SnPP to the THP-1 cells. (**J**) Immunoblotting of IKKα and HO-1 in the kidneys of mice. Data are expressed as the mean ± SEM (*n* = 4). * *p* < 0.05, ** *p* < 0.01, *** *p* < 0.001 vs. control, 0 μM group, or sham group. # *p* < 0.05, ## *p* < 0.01, ### *p* < 0.001 vs. H_2_O_2_ group or 5/6Nx group. † *p* < 0.05, ††† *p* < 0.001 vs. H_2_O_2_ + SM group.

**Figure 6 antioxidants-11-02461-f006:**
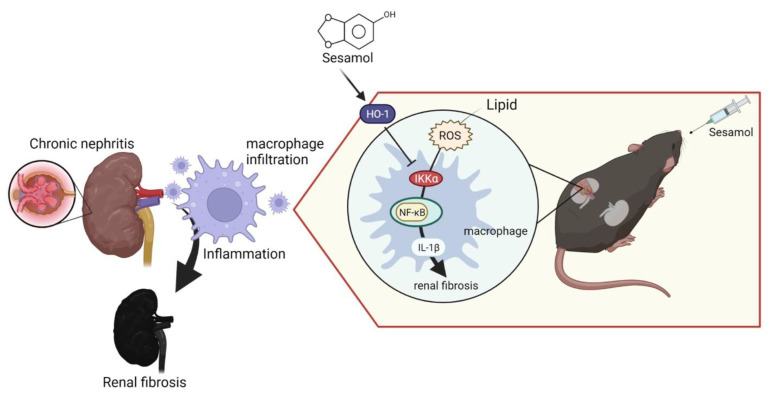
ROS-induced expression of IL-1β leads to chronic nephritis via the phosphorylation of IKKα and activation of NF-κB. Conversely, sesamol (SM) inhibits this process via HO-1.

**Table 1 antioxidants-11-02461-t001:** List of primers used for qPCR.

Gene	Forward	Reverse
Heme oxygenase-1 Interleukin 1β GAPDH	5′-AAGACTGCGTTCCTGCTCAAC-3′ 5′-GCAAGGGCTTCAGGCAGGCCGCG-3′ 5′-GAAGGTGAAGGTCGGAGTCAAC-3′	5′-AAAGCCCTACAGCA ACTGTCG-3′ 5′-GGTCATTCTCCTGGAAGGTCTGTGGGC-3′ 5′-CAGAGTTAAAAGCAGCCCTGGT-3′

## Data Availability

The data presented in this study are available in article.

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
