# Peer review of "Sesamol Attenuates Renal Inflammation and Arrests Reactive-Oxygen-Species-Mediated IL-1β Secretion via the HO-1-Induced Inhibition of the IKKα/NFκB Pathway In Vivo and In Vitro"

_antioxidants, 2022, doi:10.3390/antiox11122461_

Round 1

Reviewer 1 Report

In this manuscript, Tseng et al. evaluated the impact of sesamol on renal injury-mediated renal inflammation in nephrectomised apoE-KO mice and attempted to explore possible protective mechanism of SM in hydrogen peroxide-treated THP-1 cells. The experiments are well-conducted and the topic is interesting. However, several weaknesses need to be addressed before acceptance.

1. For this experiment, why are apoE-KO mice underwent 5/6 nephrectomy (NX) used?

2. It would be helpful if the authors could clarify how HO-1 interacts with IKKalpha.

3. In the abstract, the authors may consider to remove the abbreviation “HO-1”, because it just appears one time.

4. Moreover, the abbreviations of several words are wrong such as “OH-1” and “NF-КB”. Please check and correct it.

Reviewer 2 Report

The authors tested a murine model of chronic kidney disease (CKD) for the therapeutic potential of sesamol (SM)out of sesame seeds: The NFkB pathway was identified as the key player in CKD-mediated inflammation and affected by the treatment with SM. THP-1 cells were used as in vitro model of inflammatory cells  / macrophages and the effect of SM on the NFkB pathway was tested. The manuscript would greatly benefit from addressing the following questions or critiques.

1.       Figure 4A unit of H2O2 is missing and it is not included in the method section. I assume it is µM but this should be clarified. The justification for the 5µM of H2O2 used in the following experiments is missing in the text. The meaning of Figure 4A and 4B will be enhanced by that.

2.       Figure 4E: why is not only p-IKKa but also total IKKa induced / elevated in H2O2? Same result in Figure 5I. Any feedforward loop known on the transcriptional level that should be included in the citations? The authors should explain this in more detail.

3.       The siRNA for IKKa with 10nM final concentration seems to be very low efficient in the THP-1 cells and only reach about 50% reduction in expression of IKKa. If not done yet the authors should try to achieve a better KD of IKKa with 20nM or longer incubation time. 12h might be too short to see the KD sufficiently and many protocols include a time line of 24h to 48h post transfection. Another option is the change of the transfection reagent. The transfection reagent and manufacturer of the siRNA are not mentioned in the methods and should be provided.

4.       Figure 5C should also include the total levels of p65 NFkB and IkBa and the quality of the blots is far less than all the other blots shown in the manuscript. This figure panel should be updated with better blots.

5.       Figure 5G: The effect of SM on IL-1ß induced by H2O2 is not fully abolished as stated in the main text. The increase in IL.-1ß mRNA is reduced by maximum 20% as shown in the graph 5G. This should be rephrased in the text.

6.       Is there any function of HO-2 in macrophages as the non-inducible enzyme? What about the non-canonical NFkB pathway in this model? The authors should make at least a discuss this in the manuscript.

7.       Statistics using the students t-test is inappropriate for multiple group sizes as the usual 4 groups in the graphs. An ANOVA with post-hoc analysis would be the correct statistics to carry out and the authors should test this.

8.       In the long-term perspective of the therapeutic value of SM that authors could elaborate more in the does and daily uptake. Are the 8 weeks in the model representative of a long-term treatment of CKD and do the authors see SM as single treatment option or in combination with other anti-inflammatory drugs?
